# Resilience throughout COVID-19: Unmasking the realities of COVID-19 and vaccination facilitators, barriers, and attitudes among Black Canadians

Obidimma Ezezika[1,2]*, Toluwalope Adedugbe[3], Isaac Jonas[3], Meron Mengistu[1], Tatyana Graham[1], Bethelehem Girmay[1], Yanaminah Thullah[3], Chris Thompson[3]

1 Global Health & Innovation Lab, School of Health Studies, Faculty of Health Sciences, University of Western Ontario, London, Canada, 2 Department of Health and Society, University of Toronto Scarborough, Toronto, Canada, 3 Federation of Black Canadians, Canada

* oezezika@uwo.ca

**Data Availability Statement:** All relevant data are within the manuscript and its Supporting information files.

## Abstract

Black communities have suffered disproportionately higher numbers of COVID-19 cases and deaths in Canada. Recognizing the significance of supporting targeted strategies with vulnerable populations extends beyond the COVID-19 pandemic, as it addresses long-standing health disparities and promotes equitable access to healthcare. The present study investigated 1) experiences with COVID-19, 2) COVID-19's impact, and 3) factors that have influenced COVID-19 vaccine acceptance and uptake among stakeholders and partners from the Federation of Black Canadians' (FBC). We conducted semi-structured interviews with 130 individuals and four focus groups with FBC, including stakeholders and partners, between November 2021 and June 2022. The semi-structured interviews and focus group discussions were conducted virtually over Zoom and lasted about 45 minutes each. Conversations from interviews and focus groups were transcribed and coded professionally using team-based methods. Themes were developed using an inductive-deductive approach and defined through consensus. The deductive approach was based on Consolidated Framework for Implementation Research (CFIR) domains and constructs. First, regarding experiences with COVID-19, 36 codes were identified and mapped onto 13 themes. Prominent themes included 39 participants who experienced highly severe COVID-19 infections, 25 who experienced stigma, and 18 who reported long recovery times. Second, COVID-19 elicited lifestyle changes, with 23 themes emerging from 62 codes. As many as 97 participants expressed feelings of isolation, while 63 reported restricted mobility. Finally, participants discussed determinants that influenced their vaccination decisions, in which 46 barriers and four facilitators were identified and mapped onto nine overarching themes. Themes around the CFIR domains *Individuals*, *Inner Setting*, and *Outer Setting* were most prominent concerning vaccine adoption. As for barriers associated with the constructs *limited available resources* and *low motivation*, 55 (41%) and 46 (34%) of participants, respectively, mentioned them most frequently. Other frequently mentioned barriers to COVID-19 vaccines fell under the construct *policies & laws*, e.g., vaccine mandates as a condition of employment.

**Funding:** This study has received funding from the Public Health Agency of Canada; 2021-HQ-000156. The role of designing this study, data analysis, interpretation of data and writing the manuscript does not involve funders.

**Competing interests:** The authors have declared that no competing interests exist.

Overall, these findings provide a comprehensive and contextually rich understanding of pandemic experiences and impact, along with determinants that have influenced participants' vaccination decisions. Furthermore, the data revealed diverse experiences within Black communities, including severe infections, stigma, and vaccine-related challenges, highlighting the importance of targeted interventions, support, and consideration of social determinants of health in addressing these effects.

## Introduction

Several extant studies have examined COVID-19's impact on Black Canadians, particularly their disproportionate health outcomes compared with the general population [1–5]. This does not come as a surprise, given that pre-existing healthcare, economic, social, and political inequities only have worsened for Black Canadians throughout the pandemic [1]. Municipal- and provincial-level analyses have highlighted worse health outcomes among Black Canadians compared with most other ethno-racial groups, demonstrating COVID-19's uneven impact [3, 5]. A City of Toronto study revealed that people of color (POC) accounted for 83% of reported COVID-19 cases, while comprising only half of Toronto's population [2].

Moreover, the current literature has examined complex factors that have influenced vaccine uptake and worsened COVID-19 outcomes among Black Canadians. Extant studies examining Black Canadians' experiences with COVID-19 have revealed a distressing connection between police brutality and higher COVID-19 rates, compounded by socioeconomic disadvantages and being visible minorities [6]. In the United States, Black and Hispanic/Latinx populations have experienced significantly higher COVID-19 cases, hospitalizations, and deaths [7]. Similarly, in Canada, COVID-19 disproportionately has affected vulnerable groups comprising seniors, low-income earners, new Canadians, and visible minorities, including the Black population, particularly those with language barriers [7]. Relating to COVID-19's psychological impacts, the pandemic exacerbated mental health inequities in Black communities due to the economic precarity of frontline workers with limited income alternatives [8]. Across the literature, factors influencing vaccine uptake in Black communities have been linked to medical distrust, structural racism, disinformation, and socioeconomic impacts, emphasizing the need for targeted efforts to build trust and improve vaccine acceptance [6, 8, 9]. Overall, the pandemic has magnified existing social and economic inequalities, thereby exacerbating this group's vulnerabilities.

Our study aimed to complement the current literature by providing more generalizable data, as 130 participants from across Canada's provinces were included in the study, representing diverse ethnic groups. By including accounts of Black communities across Canada in our study, we attempted to gain a deeper understanding of the factors encouraging and discouraging vaccine uptake nationwide, while also aiming to contextualize pandemic experiences and impact. The study objectives were to: (1) assess recovery from COVID-19 among Black individuals; (2) examine COVID-19's impact; and (3) identify the barriers, facilitators, and attitudes associated with vaccines.

## Methods

### Participant selection criteria

The study's participants were recruited based on several inclusion and exclusion criteria, as delineated in this study's protocols [10]. This study's participants had to identify as Black

Canadians or individuals of African descent (including immigrants), be at least 18 years old, reside in Canada, and speak either English or French well enough to provide informed consent. The participants were recruited initially from the Federation of Black Canadians' (FBC) mailing database of individuals who had participated in previous FBC projects and/or were stakeholders with FBC. After this, we used snowball sampling, in which those FBC members initially recruited suggested others from their communities. Those suggested were then invited to participate in the study. Altogether, 1,188 individuals were identified as potential stakeholders and contacted for interviews. Of the 1,188, 206 responded to our request to participate, with 197 meeting the eligibility criteria, out of which, 185 signed up for an interview through the FBC web portal, then were contacted by our team to schedule an interview/focus group session. In the end, 130 stakeholders agreed to participate in the study and were interviewed.

## Data collection

The University of Toronto's Social Sciences, Humanities, and Education Research Ethics Board (No. 41585) approved this study. Data were collected through semi-structured interviews and focus groups from November 18, 2021 until June 08, 2022 with participants who met the eligibility criteria.

The research team obtained informed consent from participants first by explaining the purpose of the project, as well as the roles, risks, and benefits of participation, and answering any questions or concerns. Individuals who agreed to participate in interviews signed the consent form and submitted it to the research team for filing before interviews commenced. Moreover, the research team re-iterated the voluntary nature of participation, and that participants had the choice to withdraw from the study at any point. The interviews were conducted first using interview guides, and took place virtually over Zoom, each lasting approximately 45 minutes.

After the completion of interviews, participants who also agreed to take part in focus groups were provided with the consent form, which were reviewed and signed. Before focus groups begun, the research team re-iterated the project's aim, rationale, as well as the expected roles, risks and benefits of participation. Moreover, the research team notified participants that they have the choice to withdraw from the study at any point. Four focus groups, one for each province (Ontario, British Columbia, Alberta, and Quebec), were conducted, with four to seven participants in each group. The interview guide questions were utilized to conduct the focus groups, which were held virtually via Zoom and lasted around 45 minutes. Compensation of $30 was offered to participants upon completion of the semi-structured interviews and focus groups.

## Data analysis

NVivo was the primary qualitative analysis tool used. Two team members trained in using NVivo software independently coded the interview transcripts. They met 20 times, with each meeting lasting between one to two hours, to talk through coding differences and create a consensus template. The transcripts from the semi-structured interviews and focus groups were analyzed together. As an additional coding reliability check, one of the team members, an experienced qualitative researcher with extensive experience in applying and using determinant frameworks in implementation research, coded a subset of interviews and resolved any conflicts arising from the codes identified. The codes with representative quotes were organized into tables as illustrated in S1 Table. All codes, including the barriers to and facilitators of COVID-19 vaccines, were grouped into broader themes.

## CFIR application

The Consolidated Framework for Implementation Research (CFIR) is a conceptual framework created to guide the systematic assessment of factors that influence interventions' implementation and effectiveness [11]. Incorporating the CFIR during the analysis and synthesis phase was highly beneficial, as integrating a conceptual framework increased the study results' generalizability and interpretability. The CFIR codebook is quite comprehensive and described 48 constructs and 19 subconstructs across five broad domains [12]. The CFIR helped triangulate information and allowed us to conceptualize barriers and facilitators in a more organized manner. The application of the CFIR was assessed by reviewing each barrier and facilitator, and coding them based on one of the CFIR framework's constructs. Two team members independently coded each facilitator and barrier based on the CFIR framework, then held discussions to identify and resolve any disagreements between the two sets of data coded. The total interrater reliability score was 83.8% for the barriers and facilitators relating to experiences with vaccines. A third team member was brought in to resolve any remaining conflicts between the first two team members when coding the collected data by determining which CFIR code for each barrier and facilitator would be the most appropriate based on the CFIR definitions.

## Results

### Participants' sociodemographic characteristics

The individuals who participated in the study ranged in age from 20–89. From the sample size of 130, 34 participants (26.1%) were between ages 30 and 39. In addition, while 78 participants identified as female (60%), 49 (37.7%) identified as male. For this study's purposes, we limited our scope of analysis to sex (female or male) rather than gender, which encompasses a wide range of expressions of identities beyond biological attributes. Regarding ethnicity, 46 participants identified as "Black" (35.4%), while 37 (28.5%) identified as "African." Participants from all provinces were included, such as those who resided in Ontario (42, 32.3%), followed by British Columbia (30, 23.1%), Alberta (17, 13.1%), and Quebec (12, 9.2%). Furthermore, 97 (74.6%) reported being employed, 13 (10%) reported being unemployed, eight (6.2%) reported being retired, and seven (5.4%) were self-employed. Regarding highest education level, 46 participants had a master's degree (35.4%), followed by 41 with a bachelor's degree (31.5%). Finally, 64 (49.2%) reported being single, and 46 (35.4%) reported being married. The participants' sociodemographic characteristics are provided in Table 1.

### Experiences with COVID-19

*Experiences with COVID-19* described themes related to several encounters with COVID-19 infection. Altogether, 13 themes emerged from this topic area and can be found in Table 2.

Three prominent themes surfaced among the 13 that emerged. First, 39 participants described experiencing *high severity of COVID-19 infection*. Second, 23 participants mentioned *experiences of stigma*. Third, 18 participants cited *long recovery* as the third most prominent theme.

Under the *high severity of COVID-19* theme, the most prominent factors, cited by 12 and 11 participants, included family member(s) getting COVID, and severe symptoms and difficult recovery, respectively. Most participants who experienced high severity of COVID-19 were between ages 20–29 (45%) and were employed (72%). Furthermore, six participants claimed that (a) family member(s) hospitalized for COVID-19 was a significant factor. As one

**Table 1. Semi-structured interview participants' sociodemographic characteristics (n = 130).**

| Demographics | | |
|---|---|---|
| *Age* | Number of participants | Percentage of participants (%) |
| 20–29 | 33 | 25.4 |
| 30–39 | 34 | 26.1 |
| 40–49 | 26 | 20.0 |
| 50–59 | 17 | 13.1 |
| 60–69 | 11 | 8.5 |
| 70–79 | 3 | 2.3 |
| 80–89 | 1 | 0.8 |
| No answer | 5 | 3.8 |
| *Employment Status* | | |
| Employed | 97 | 74.6 |
| Retired | 8 | 6.2 |
| Self-employed | 7 | 5.4 |
| Unemployed | 13 | 10.0 |
| No answer | 5 | 3.8 |
| *Ethnicity* | | |
| African | 37 | 28.5 |
| African American | 1 | 0.8 |
| African Canadian | 3 | 2.3 |
| Afro-Caribbean | 7 | 5.4 |
| Afro-indigenous | 1 | 0.8 |
| Black | 46 | 35.4 |
| Black African | 31 | 23.8 |
| Black Antiguan | 1 | 0.8 |
| Black Canadian | 5 | 3.8 |
| Black Caribbean | 6 | 4.6 |
| Cameroonian | 1 | 0.8 |
| Canadian-Caribbean | 2 | 1.5 |
| Caribbean | 2 | 1.5 |
| Congolese | 1 | 0.8 |
| East African | 2 | 1.5 |
| Ethiopian | 1 | 0.8 |
| Ghanaian | 1 | 0.8 |
| Guyanese | 1 | 0.8 |
| Haitian | 1 | 0.8 |
| Indigenous | 1 | 0.8 |
| Jamaican | 5 | 3.8 |
| Jamaican Black | 4 | 3.1 |
| Jamaican Canadian | 2 | 1.5 |
| Kenyan | 2 | 1.5 |
| Nigerian | 8 | 6.2 |
| Rwandan | 2 | 1.5 |
| Senegalese African Korean | 1 | 0.8 |
| Somali | 1 | 0.8 |
| Tanzanian | 1 | 0.8 |
| Trinidadian | 1 | 0.8 |
| Ugandan | 1 | 0.8 |

*(Continued)*

**Table 1.** (Continued)

| Demographics | | |
|---|---|---|
| West Indian | 1 | 0.8 |
| Zimbabwean | 2 | 1.5 |
| *Sex* | | |
| Female | 78 | 60.0 |
| Male | 49 | 37.7 |
| No answer | 3 | 2.3 |
| *Highest Level of Education* | | |
| Associate's degree | 2 | 1.5 |
| Bachelor's degree | 41 | 31.5 |
| College degree (i.e. Community college) | 10 | 7.7 |
| Doctorate degree | 9 | 6.9 |
| High school | 4 | 3.1 |
| Law school | 1 | 0.8 |
| Master's degree | 46 | 35.4 |
| No answer | 8 | 6.2 |
| Postgraduate certificate (diploma) | 6 | 4.6 |
| Postsecondary | 1 | 0.8 |
| Postsecondary certificate | 1 | 0.8 |
| Postsecondary degree | 1 | 0.8 |
| *Marital Status* | | |
| Common law | 8 | 6.2 |
| Divorced | 8 | 6.2 |
| Married | 46 | 35.4 |
| No answer | 3 | 2.3 |
| Separated | 1 | 0.8 |
| Single | 64 | 49.2 |
| *Province of Residence* | | |
| No answer | 4 | 3.1 |
| Alberta | 17 | 13.1 |
| British Columbia | 30 | 23.1 |
| Manitoba | 7 | 5.4 |
| New Brunswick | 11 | 8.5 |
| Newfoundland | 1 | 0.8 |
| Northwest Territories | 1 | 0.8 |
| Nova Scotia | 3 | 2.3 |
| Nunavut | 1 | 0.8 |
| Ontario | 42 | 32.3 |
| Prince Edward Island | 1 | 0.8 |
| Quebec | 12 | 9.2 |

*Note.* The total number of participants (n = 130) represents the 130 participants from the semi-structured interviews.

participant described it, "*My father, he was affected early, April 2020, and he was at the hospital for 11 days*" (Participant 32).

Altogether, 13 participants experienced stigma during the COVID-19 pandemic, e.g., one participant from Quebec stated the following: "*The popular thought of saying the omicron is from South Africa, and as a Canadian-born Senegalese, it's hard for me because they look at me*

**Table 2. Overarching thematic areas identified from the semi-structured interview and focus group transcripts and thematic definitions for topic area: Experiences with COVID-19 (n = 134).**

| | Experiences With COVID-19 | | | |
|---|---|---|---|---|
| | **Themes** | **Descriptions** | **Number of codes** | **Code names** |
| 1. | High severity of COVID-19 infection | *COVID-19 infection severely impacted health, increasing the likelihood of death* | 9 | Family member(s) got COVID (12)<br>Severe symptoms and difficult recovery (11)<br>Family member(s) hospitalized (6)<br>Required oxygen therapy (3)<br>Someone close got hospitalized (2)<br>Stayed in ICU (2)<br>Friends(s) hospitalized (1)<br>Scary experience (1)<br>Hospitalized due to severity, required oxygen (ON) |
| 2. | Experiences of stigma | *Experiences of stigma after being infected with COVID-19.* | 5 | Experienced stigma (13)<br>Someone close got stigmatized (5)<br>Family members experienced stigma (3)<br>Stigma for getting COVID after not getting vaccine (1)<br>Family member ostracized (1) |
| 3. | Long recovery | *Extended duration of COVID-19 infection lasting for over two weeks.* | 4 | Over two weeks to recover (8, QC)<br>Long recovery (5)<br>Over four weeks before testing negative (2)<br>Slow recovery (1) |
| 4. | Short recovery | *Duration of infection lasting from a few days to less than two weeks after being infected with COVID-19* | 3 | Fine after a few days (6)<br>Recovered quickly (6)<br>Difficult for some days (4) |
| 5. | Average recovery | *Duration of infection lasting the average or normal two-week period after being infected with COVID-19* | 1 | Recovered within two weeks (17, AB, QC) |
| 6. | Family passing away | *Family members or friends died due to COVID-19 infection.* | 2 | Family member(s) passed away due to COVID (7, QC)<br>Someone close died (3) |
| 7. | Lacked emotional and physical support | *Individual did not receive emotional and/or physical support from family or friends while recovering from COVID-19 infection.* | 2 | Lacked support (6)<br>Person didn't get support (1) |
| 8. | Not needing support | *Expressed not needing support during COVID-19 recovery.* | 1 | Didn't need support (2) |
| 9. | Social support from family and friends | *Individual received support while recovering from COVID-19 infection.* | 1 | Family or friends provided support (QC, AB) |
| 10. | Family members lacked support | *Family members lacked support during COVID-19 recovery.* | 1 | Family lacked support (2) |
| 11. | Felt guilty for getting COVID-19 | *Felt guilty for getting COVID-19.* | 1 | Felt guilty for getting COVID-19 (ON) |
| 12. | Inadequate financial support | *Individual did not receive adequate financial support.* | 2 | Inadequate financial support due to employment status (2)<br>Lack of financial support due to immigration status (1) |
| 13. | Lacked informational support | *Individual did not receive informational support while recovering from COVID-19 infection.* | 1 | Lacked informational support (1) |

*Note*. The total number of participants (n = 134) represents the 130 participants from the semi-structured interviews, plus the four focus groups. The number in brackets following each code name indicates the number of participants who mentioned the code. The letters in brackets following each code name indicate the focus groups and represent the provinces in which the focus groups were conducted, including Ontario (ON), Quebec (QC), Alberta (AB), and British Columbia (BC).

*as guilty when they see me*" (Participant 26). Of the 13 participants who described experiencing stigma, 77% identified as female, and 54% had a master's degree. Furthermore, five participants mentioned that someone close was stigmatized in the course of a COVID-19 infection.

Finally, under the theme *long recovery*, nine participants required over two weeks to recover from COVID-19. Participants between ages 50–59 and those residing in Ontario described requiring over two weeks to recover more frequently than other participants. Similarly, five reported having a long recovery when overcoming a COVID-19 infection. As one participant noted, *"Even the ones that didn't go into the ICU took months, and I don't think they can say they've fully recovered"* (Participant 20).

Other themes related to *experiences with COVID-19* included *lacked emotional and physical support*, which highlighted the experiences of six participants who did not receive emotional or physical support from family or friends while recovering from COVID-19. Regarding *inadequate financial support*, three participants reported that COVID-19 negatively impacted their financial status due to employment and immigration status.

## Experiences with COVID-19 vaccines

*Experiences with COVID-19 vaccines* described the positive and negative factors that participants deemed influential in relation to COVID-19 vaccines and their uptake. In this topic area, nine themes emerged from the analyses, which can be found in Table 3. Participants described 46 barriers and four facilitators related to COVID-19 vaccines.

Among the nine themes that emerged from the analyses, two were prominent. Altogether, 45 participants mentioned factors related to *inaccessibility and unavailability of vaccines*. Furthermore, 27 participants cited *lack of information and communication*.

*Inaccessibility and unavailability of vaccines* described the limited ability to obtain COVID-19 vaccines due to barriers related to accessibility and availability. Participants described factors that impeded their ability to receive the vaccines, including infrastructural issues and vaccine deployment strategies. Within this theme, three factors were prominent: long lines to get the vaccine; a lack of vaccine facilities; and vaccine locations' inaccessibility. One participant described their experiences with long lines, stating that *"you would go out and spend a number of hours to be able to get the shot. So that was a major barrier for me"* (Participant 128).

*Lack of information and communication* described the limited access to accurate vaccine information and the challenges in obtaining vaccine-related information. Participants described misinformation's dominance in the absence of information from government sources, resulting in inconsistencies related to COVID-19 vaccines. The most frequently mentioned factor related to this theme was misinformation or inconsistent information surrounding COVID-19 vaccines, e.g., one participant highlighted that "*there was a lot of misinformation. So much so that I may have delayed taking the vaccine so that I could get a specific vaccine*" (Participant 122). Moreover, participants mentioned a lack of information regarding how to receive the vaccines six times, which was deemed a significant barrier.

Other notable themes included *feeling coerced to get the vaccine, fear of nonvaccination consequences*, and *negative experiences with vaccine mandates*. With specific reference to racialized individuals' experiences, one participant stated the following: "*I have had more issues with the vaccine passport system in terms of experiencing anti-Black racism. Establishments single me out to show my papers proving vaccination when I am the only Black person in the store*" (Participant within the Ontario focus group).

The CFIR results presented in Table 4 demonstrate that participants described more negative than positive factors concerning experiences with COVID-19 vaccines. Barriers related to the construct *available resources* were dominant and mentioned by 55 (41%) participants, who

**Table 3. Overarching thematic areas identified from the semi-structured interview and focus group transcripts and thematic definitions for topic area: Experiences with COVID vaccines (n = 134).**

| | | Experiences With COVID-19 Vaccines | | |
|---|---|---|---|---|
| | **Themes** | **Descriptions** | **Number of codes** | **Code names** |
| 1. | Inaccessibility and unavailability of vaccines | *Limited ability to obtain COVID-19 vaccines due to accessibility and availability issues.* | 15 | Long lines to get vaccine (9)<br>Inaccessibility to vaccine locations (7)<br>Lack of access (6)<br>Better technology or booking system (6)<br>Door-to-door vaccine service (3)<br>Lack of vaccine variety options (3)<br>Difficulty accessing vaccine (2)<br>Location decreased access (2)<br>Accessibly for those with a disability (1)<br>First-come-first-serve (1)<br>Unavailable for age group (1)<br>Lack of health coverage (1)<br>Lacked access to desired vaccine (AB)<br>Feels restricted from getting vaccine due to age (BC)<br>Lack of accommodation (QC) |
| 2. | Lack of Information and communication | *Limited access to accurate vaccine information and experienced challenges obtaining vaccine-related information* | 5 | Misinformation or inconsistent information (12)<br>Lack of information (6)<br>Lack of knowledge (4)<br>Lack of information from government (3)<br>Misinformed or confused about vaccine (AB, QC) |
| 3. | Feeling coerced to receive vaccine | *Any expression or disfavor toward vaccine deployment strategies, politicization, and public health approaches related to COVID-19 vaccines.* | 8 | Disfavor of coercive tactics (10)<br>Felt forced to get vaccine (5)<br>Feels it's rushed (2)<br>Vaccinated because it was required (BC, QC)<br>Dislikes vaccine being politicized (1)<br>Feels the government could be more compassionate (QC)<br>Stressed about taking vaccine to keep employment (QC)<br>Vaccinated in secret due to stigma (ON) |
| 4. | Negative beliefs about the vaccine | *Expression of opposition, resistance, and cynicism toward COVID-19 vaccines.* | 11 | Believes it's controlling and harmful (3)<br>Believes it's horrible (2)<br>Believes it's a necessary evil (1)<br>Believes it's a tool to depopulate (1)<br>Believes it's an experiment (1)<br>Believes unvaccinated are brave (BC)<br>Regrets taking vaccine (BC)<br>Believes vaccine is useless (4)<br>Doesn't trust it (3)<br>Believes it's not a solution (2)<br>Would not get vaccines (BC) |
| 5. | Vaccine skepticism and reservation | *Expression of reservations and skepticism regarding COVID-19 vaccines.* | 4 | Skeptical (9)<br>Wary of constant vaccination (5)<br>Reserved about vaccine (BC)<br>Fearful of long-term impact (BC) |

(*Continued*)

**Table 3.** (Continued)

| | Themes | Descriptions | Number of codes | Code names |
|---|---|---|---|---|
| | | **Experiences With COVID-19 Vaccines** | | |
| 6. | Comfortable with vaccine | *Feelings of physical and mental ease toward COVID-19 vaccines.* | 1 | Comfortable with vaccine (AB, BC, ON, QC) |
| 7. | Beliefs about vaccine immunity | *Beliefs about the immunity provided by vaccines.* | 2 | Two-dose vaccines provide more immunity (BC, QC) Believes vaccine protects the vulnerable (ON) |
| 8. | Fear of nonvaccination consequences | *Expression of uncertainty toward COVID-19, thereby opting to take vaccines to prevent negative consequences.* | 1 | Fear due to uncertainty about COVID (BC, QC) |
| 9. | Negative experiences with vaccine mandates | *Belief that government- mandated vaccine passports fueled racially targeted screening of Black individuals.* | 1 | Feels racially targeted upon vaccine passport request (ON) |

*Note*. The total number of participants (n = 134) represents the 130 participants from the semi-structured interviews, plus the four focus groups. The number in brackets following each code name indicates the number of participants who mentioned the code. The letters in brackets following each code name indicate the focus groups and represent the provinces in which the focus groups were conducted, including Ontario (ON), Quebec (QC), Alberta (AB), and British Columbia (BC).

**Table 4.** **Frequency of cited Consolidated Framework for Implementation Research (CFIR) constructs on barriers and facilitators to experiences with vaccine (n = 134).**

| | **COVID-19 Vaccines** | |
|---|---|---|
| **CFIR domains (n = 5) and constructs (n = 7)** | **Barrier n (%) of interviews** | **Facilitator n (%) of interviews** |
| *I. Innovation Domain* | | |
| *No facilitators or barriers were noted for the following constructs: Innovation Source; Innovation Evidence-Base; Innovation Relative Advantage; Innovation Adaptability; Innovation Trialability; and Innovation Complexity* | | |
| Innovation Design | 2 (1.5%) | 0 |
| *II. Outer Setting Domain* | | |
| *No facilitators or barriers were noted for the following constructs: Critical Incidents; Local Attitudes; Local Conditions; Partnerships and Connections; Financing; and External Pressure (including subconstructs Societal Pressure, Market Pressure, and Performance-Measurement Pressure)* | | |
| Policies & Laws | 12 (9%) | 2 (1.5%) |
| *III. Inner Setting Domain* | | |
| *No facilitators or barriers were noted for the following constructs: Structural Characteristics (including Physical Infrastructure, Information Technology Infrastructure, and Work Infrastructure); Relational Connections; Communications; Culture (Human Equality-Centeredness, Deliverer-Centeredness, and Learning-Centeredness); Tension for Change; Compatibility; Relative Priority; Incentive Systems; Mission Alignment; Available Resources (Funding); and Access to Knowledge and Information* | | |
| Culture: Recipient-Centeredness | 1 (0.7%) | 0 |
| Available Resources | 55 (41%) | 0 |
| *IV. Individuals' Domain: Roles Subdomain* | | |
| *No facilitators or barriers were noted for all constructs under this subdomain* | | |
| *IV. Individuals' Domain: Characteristics Subdomain* | | |
| Capability | 6 (4%) | 2 (1.5%) |
| Motivation | 46 (34%) | 7 (5%) |
| *V. Implementation Process Domain* | | |
| *No facilitators or barriers were noted for all constructs; Teaming; Assessing Needs (Innovation Deliverers); Assessing Context; Planning; Tailoring Strategies; Engaging (Innovation Deliverers, Innovation Recipients); Doing; Reflecting and Evaluating (Implementation, Innovation); and Adapting* | | |
| Assessing Needs: Innovation Recipients | 9 (7%) | 0 |

*Note*. The total number of participants (n = 134) represents the 130 participants from the semi-structured interviews, plus the four focus groups. The total number of participants (n = 134) represents the 130 participants from the semi-structured interviews, plus the four focus groups.

described barriers such as inaccessibility to vaccine facilities. Other frequently mentioned barriers and challenges related to experiences with COVID-19 vaccines fell under the constructs *motivation* (mentioned by 46 participants, 34%), which included lack of commitment to vaccine uptake, and *policies & laws*, e.g., vaccine mandates as a condition of employment (mentioned by 12 participants, or 9%).

Participants cited three facilitators falling under the constructs *motivation* (seven participants, 5%) which included positive beliefs about the vaccine that encourage uptake; *policies & laws*, e.g., vaccine mandates as a condition of employment, which in this case, encouraged vaccine uptake (two participants, 1.5%); and *capability* (two participants, 1.5%), which described participants' competence and knowledge surrounding vaccine uptake.

## COVID-19's impact

*COVID-19's impact* touched on lifestyle changes that COVID-19 elicited. This topic area comprised 23 themes (Table 5).

Participants expressed *feelings of isolation* a total of 105 times, which described the situation of being physically separated from family, friends, or society due to the COVID-19 pandemic. The following theme, *restricted mobility*, described the limited ability to move physically from one location to another, and 81 participants cited it.

Under the *feelings of isolation* theme, 45 participants expressed feeling socially isolated due to the COVID-19 pandemic. Furthermore, 86% of these participants were employed, and those identifying as single (48%) followed by married (36%) described this feeling of social isolation. Furthermore, 33 participants were unable to visit family or friends. One participant described the pandemic's impact as follows: *"There have been difficult periods during COVID-19 because of isolation, because of not being able to see friends and family or travel, or go to work, or just live the life I am used to"* (Participant 42).

Moreover, as a result of the COVID-19 pandemic, 20 participants were unable to travel due to restricted mobility. One expressed that because of the pandemic, *"We are not as mobile as we normally are. We could not have gone away or taken a vacation or anything because we tried to stay put"* (Participant 9). Participants who described restricted mobility and its challenges mostly were those residing in Ontario (44%) and those identifying as female (61%).

This topic area also presented other themes. *Work opportunity* drew attention to positive and/or negative work-related outcomes impacted by the COVID-19 pandemic. Furthermore, *challenges adjusting to remote school, work, and events* described experiences utilizing technology to access several spaces during the COVID-19 pandemic. *Concerns related to well-being and access to basic needs* point to the importance of recognizing the concerns surrounding COVID-19 ideas and outcomes.

## Discussion

These findings from the semi-structured interviews and focus groups provide unique and valuable insights into Black populations' experiences and attitudes across Canada. These qualitative methods' objectives were to uncover COVID-19 experiences and impacts, along with barriers to vaccine uptake, and the results provide contextually rich data on Black populations' perspectives across Canada.

Regarding experiences with COVID-19, physical and social consequences seemed to impact participants the most. This study's participants were affected profoundly, both physically and socially, with some enduring severe symptoms, hospitalization, and extended infections, while others faced stigma, emotional strain, financial struggles, and a lack of support from family and friends. The high severity of COVID-19 infection was the most frequently mentioned

**Table 5. Overarching thematic areas identified from the semi-structured interview and focus group transcripts and thematic definitions for topic area: COVID-19's Impact (n = 134).**

| | Themes | Description | Number of codes | Code names |
|---|---|---|---|---|
| | | **COVID-19's Impact** | | |
| 1. | Feelings of isolation | *Being physically separated from family, friends, or society due to the COVID-19 pandemic* | 8 | Socially isolated (42, BC, ON, AB)<br>Unable to visit family and/or friends (31, BC, ON)<br>Limited interactions (17)<br>Unable to go to church (6)<br>Unable to have social gatherings (1)<br>Confined to home (1)<br>Created community divide (1)<br>Lost friendship (ON) |
| 2. | Restricted mobility | *Limited ability to move physically from one location to another.* | 7 | Unable to travel (18, QC, AB)<br>Unable to visit family and/or friends (31, BC, ON)<br>Restricted mobility (10)<br>Difficulty traveling (8)<br>Restricted traveling (7)<br>Increased cost to travel (2)<br>Stopped going out (1) |
| 3. | Poor mental health outcomes | *Negative mental health effect experienced.* | 1 | Negatively affected mental health (21) |
| 4. | Work opportunity | *Received work opportunities or had difficulty finding a job during the COVID-19 pandemic.* | 5 | Difficulty getting a job (13, BC, ON)<br>Received work opportunity (QC, BC)<br>Business was lost (1)<br>Stressed about taking vaccine to keep employment (QC)<br>Difficulty finding affordable housing (1) |
| 5. | Challenges adjusting to remote school, work, and funeral | *Expressed challenges and negative feelings toward utilizing online resources (e.g., Zoom) for work, school, or funerals.* | 5 | Adjusting to work-at-home life (9)<br>Challenges adjusting to remote work or school (BC, ON)<br>Mourned loss on Zoom (1)<br>Remote-learning negative outcomes (QC)<br>Working remotely increased convenience (QC) |
| 6. | Concerns related to well-being and access to basic needs | *Expressed concerns about others' well-being and getting infected with COVID-19.* | 3 | Concerned about getting COVID (8)<br>Concerned for others' well-being (4)<br>Concerned about access to basic needs (1) |
| 7. | Changes to social life | *Changes in social health due to COVID-19 pandemic.* | 5 | Altered interaction with students (1)<br>Felt lonely (QC)<br>Limited access to support network (QC)<br>Unable to celebrate events (BC)<br>Social skills regressed (ON) |
| 8. | Increased stress | *Experienced stress or additional stressors due to COVID-19 pandemic.* | 2 | Stressed (4)<br>Added stressors (3) |
| 9. | Disruption of productivity, routine, and life plans | *Changes in routine or productivity.* | 3 | Altered routine and activities (4)<br>Decreased productivity (1)<br>Interfered with life plans (BC) |
| 10. | Work structure changes | *Experienced changes in job structure.* | 1 | Change in job structure (6) |
| 11. | Mental peace and social benefits | *Expressed mental peace, increased time to spend with family members, and decreased microaggressions.* | 4 | Found time to spend with family (BC, QC)<br>Felt mentally at peace (QC)<br>Fewer microaggressions or inequities experienced (QC)<br>Received leisure time (QC) |

(*Continued*)

**Table 5.** (Continued)

| | Themes | Description | Number of codes | Code names |
|---|---|---|---|---|
| | | **COVID-19's Impact** | | |
| 12. | Resource insecurity | *Uncertainty or worry about lack of essential or basic human needs due to COVID-19 pandemic.* | 4 | Decreased income (ON, AB)<br>Food insecurity (ON)<br>Unable to afford Internet to schedule vaccine (QC)<br>Difficulty finding affordable housing (1) |
| 13. | Increased anxiety | *Experience of increased anxiety due to COVID-19 pandemic.* | 2 | Increased anxiety (QC, ON, BC, AB)<br>Worried about community (QC) |
| 14. | Challenges parenting | *Expressed challenges related to parenting during the COVID-19 pandemic.* | 3 | Challenging as a single parent (1, QC)<br>Difficulty working remotely with kids (BC)<br>Difficulty shopping with kids (QC) |
| 15. | Polarized opinions in the Black community | *The COVID-19 pandemic created polarized opinions within the Black community.* | 1 | Polarized opinions in Black community (QC) |
| 16. | Difficulty Adjusting post-pandemic | *Expressed difficulty adjusting post-pandemic.* | 1 | Difficulty adjusting post-pandemic (BC) |
| 17. | Experienced loss due to suicide | *Experienced loss due to suicide.* | 1 | Lost friends to suicide (QC) |
| 18. | Feelings of being targeted by the government | *Expressed feelings of being targeted by the government.* | 1 | Feels targeted by government (1) |
| 19. | Concerns of in-person classes | *Expressed stress related to in-person classes.* | 1 | Stressed about in-person classes (1) |
| 20. | Decreased physical activity | *Physiological decline due to COVID pandemic.* | 1 | Decreased physical activity (AB) |
| 21. | Relief being around community | *Expressed relief as an outcome of being around members of the Black community.* | 1 | Relief being around Black community (QC) |
| 22. | Transportation mode changes | *Mode of transportation changed due to COVID-19 pandemic.* | 1 | Altered mode of transportation (1) |
| 23. | Limited information about COVID | The initial beliefs about COVID-19 susceptibility | 1 | Limited conclusive info about COVID (QC) |

*Note*. The total number of participants (n = 134) represents the 130 participants from the semi-structured interviews, plus the four focus groups. The number in brackets following each code's name indicates the number of participants that mentioned the code. The letters in brackets following each code's name indicate the focus groups and represent the provinces in which the focus groups were conducted, including Ontario (ON), Quebec (QC), Alberta (AB), and British Columbia (BC).

theme in this topic area. In a Canadian study that included various ethno-cultural groups (White, East Asian, South Asian, Black, Southeast Asian, Arab, and others), Black Canadians were more likely to report COVID-19 symptoms (their own or those of someone they knew), more likely to say they sought treatment for COVID-19, more likely to report the worst mental health outcomes when exposed to COVID-19 and/or COVID-19-related discrimination, and nearly three times as likely to report knowing someone who had died from the virus [13]. Moreover, while Black Torontonians comprise 9% of Toronto's population, they comprised 13% of COVID-19 cases (55% higher than the rate of the rest of the population) and 16% of COVID-19 hospitalizations (364/100,000 Black people, compared with the overall rate of 206/100,000) [3]. Other studies consistently have reinforced the notion that Black communities in Canada have experienced a comparatively greater impact from COVID-19, highlighting disparities in infection rates, hospitalizations, and mortality rates when compared with other racial and ethnic groups [4, 14, 15]. Similarly, studies conducted in the United States have demonstrated that Black communities have faced a disproportionate COVID-19 burden compared with other racial and ethnic groups, thereby emphasizing the presence of systemic disparities and healthcare inequities [16–18].

The high severity of COVID-19 infection was the most frequently mentioned theme, and the social consequences, e.g., experiences of stigma and lack of emotional and social support from family and friends, also made a significant impact. As examined in the Canadian study that assessed COVID-19 outcomes among various ethno-cultural groups, Black participants reported the worst mental health outcomes when exposed to COVID-19 and/or COVID-19-related discrimination [13]. This study's findings suggest that experiences of exposure and discrimination exert differential effects across ethno-cultural groups, putting Black participants at the highest risk of mental distress and various mental health challenges compared with other ethno-cultural groups [13]. Another study reported that online COVID-19 misinformation and fear of loss of community due to split opinions on the vaccine have impacted Black Canadians' mental health significantly [19]. Social and mental health challenges often were exacerbated during the pandemic; however, Black residents reported having limited access to mental health services [19]. Our study highlights the inadequacy of current health and social support systems for Black Canadians, worsening health equity disparities during pandemics. Policymakers must prioritize innovative, community-centric approaches, including substantial investments in mental health services (e.g., digital-based interventions to increase connection), housing, and employment support [8, 20].

Our findings also revealed that participants encountered more barriers than facilitators in their experiences with COVID-19 vaccines. The first two factors that negatively influenced participants' experiences included policies/laws (i.e., vaccination mandates as a condition of employment) and available resources (i.e., inaccessible vaccine locations). Echoing our study's results, one article highlighted that confidence in and uptake of the vaccines would improve not only by communicating information about the vaccines and public health measures, but also by providing greater access to vaccines [21]. However, another study focused on the association between experiences of racial discrimination in health care and mistrust of the vaccine in the Black community. They concluded that more tools and training are needed to eliminate racism from health care systems, suggesting that more vaccines alone are insufficient [22]. The third most dominant barrier related to participants' experiences with COVID-19 vaccines was their motivation. This included beliefs that vaccines are controlling, harmful, or rushed; general distrust of vaccines; and skepticism, which 40% of participants cited. Our results highlighted various factors that influenced COVID-19 vaccination intention, including sociodemographic factors, anxiety about contracting the virus, and specific mental health concerns [23].

Notably, Black ethnicity emerged as a significant independent predictor of vaccine hesitancy compared with vaccine readiness in another study [24]. Across the general Canadian population, groups designated as visible minorities were slightly less willing to receive the COVID-19 vaccine than nonvisible minorities (74.8% and 77.7%, respectively). By comparison, 56.4% of the Black population reported being somewhat or very willing to receive a COVID-19 vaccine, which is much lower than both visible and nonvisible minorities [25]. Similar findings demonstrate that Black Canadians exhibited lower vaccine willingness compared with other groups despite facing higher risks and consequences related to COVID-19 [23]. This indicates that, while a notable lack of willingness to receive COVID-19 vaccines was observed across various racial groups, Black Canadians exhibited the lowest inclination among all racial groups. This disparity could pose a more significant challenge within these communities, as lower willingness to receive available vaccines increases the likelihood of COVID-19's impacts, as observed in our findings. However, this lower willingness to receive vaccines among Black communities in Canada reflect a myriad of reasons that are unique to their experiences as racialized minorities, including lower trust in vaccines and the healthcare system

overall, lower health literacy, underrepresentation in vaccine trials, and experiences with discrimination and systemic racism [26].

Our results indicate vaccine hesitancy in Black communities is linked not only to widespread misinformation on COVID-19 vaccines and gaps in health literacy, but also to stigma, medical distrust, and experiences of structural racism. One qualitative study found that facilitators of misinformation in the Black community included cultural/religious factors, experiences with racism in healthcare, and lack of trust in the government [27]. Expanding on these themes, another study highlighted barriers to vaccine uptake among Black communities, including anti-Black racism within health systems and society, a history of medical abuse and unethical research targeting Black communities, rapid development of vaccines, misinformation, and limited access to vaccines [23]. Even after controlling for socio-demographic differences (e.g., education and employment status, income) in their study sample, Gerretsen and colleagues identified higher vaccine hesitancy among Black, Indigenous, and Latinx populations in the US and Canada compared to White individuals [28]. This highlights the influence of historical and contemporary systemic issues contributing to mistrust in medical interventions unique to racial minorities in North America, such as the unethical Tuskegee syphilis study in the U.S. among Black American men, and the Qu'Appelle vaccine trial among First Nations children in Saskatchewan [28–30]. These factors were identified as significant challenges that contribute to lower vaccine uptake rates within this population [23].

Another significant challenge lies in the small sample sizes of current studies, thereby complicating the generalization of findings to the broader population and necessitating the compilation of results from multiple studies to construct a comprehensive understanding of vaccine hesitancy across populations [20, 31]. Despite the absence of race-based data collection in Canada, clear evidence suggests worse health outcomes and higher vaccine hesitancy among Black Canadians, emphasizing the urgent need for such data to inform timely public health responses [20]. Tailoring strategies involving Black community members, health leaders, and academics is crucial for addressing disparities, while sustaining equitable COVID-19 response policies and prioritizing race-based data collection and analysis are essential steps for building a more inclusive and responsive public health system [20, 31].

The topic area concerning COVID-19's impact revealed the most diverse findings, including social, physical, mental, and economic factors that influenced participants both positively and negatively. Of all the factors mentioned, COVID-19's social consequences made the greatest negative impact on participants. For instance, feelings of social isolation; challenges adjusting to remote school, work, and events; and changes to social life were expressed as significant COVID-19 impacts. One unique theme that also emerged from our results was the polarized COVID-19-related opinions that appeared to weaken a sense of community. For instance, misinformation-induced fear and anxiety created a rift within the Black community, as concerns over stigma related to vaccine acceptance or refusal caused a loss of community cohesion and heightened anxiety surrounding COVID-19, ultimately impeding vaccine uptake and advocacy efforts. Several other studies have reported findings similar to our results, including Black adults experiencing worsened social connections and stronger feelings of isolation; additional challenges from language barriers during isolation, making it more difficult to access information or medical help; and difficulty adjusting to remote learning for their children as a result of a lack of funding for laptops or Internet access compared with Canadians on average [32, 33].

## Limitations

This study contains several limitations that must be considered when interpreting the results. First, a potential limitation was our interviewee pool, which was limited by the number of

participants that we discovered through the FBC database, referrals, and snowball sampling. Second, there might be challenges to the findings' generalizability due to potential barriers for the target population to access technology, as the interviews were conducted virtually. Third, the interview languages were limited to English and French, therefore, individuals who could not speak either could not participate in the study. Additionally, because we solely examined sex-related factors, our findings may not fully account for gender-related nuances, which encompass a diverse range of identities beyond biological distinctions. Furthermore, the theme *experiences of stigma* under the Experiences with COVID-19 topic area might have created a biased response, as the term *stigma* was used in the question. Finally, the number of participants attributed to each theme in the results was determined by aggregating the total number of participants across all the codes within each theme. Therefore, this number could be lower if one or more participants are associated with several codes within a given theme.

## Conclusion

This study delved into COVID-19 experiences and impact among Black communities in Canada, as well as barriers to and facilitators of vaccine uptake. The findings revealed severe and varied COVID-19 experiences, including high infection severity, social stigma, and extended recovery periods. The study also highlighted the challenges concerning attitudes toward and experiences with COVID-19 vaccines, with barriers associated with CFIR constructs such as policy & laws and available resources, while individuals' motivation played dual roles as barriers and facilitators. Furthermore, the pandemic's psychological impact was evident, with feelings of isolation and restricted mobility reported. These findings underscore the need for tailored interventions and ongoing support to address COVID-19's multifaceted effects on Black individuals and communities in Canada, while also emphasizing the significance of considering the social determinants of health and equity in promoting vaccine acceptance and addressing the pandemic's broader impact on mental well-being and community resilience.

## Supporting information

**S1 Data. Interview guide.**
(DOCX)

**S1 Table. Codes and their representative quotes related to experience with COVID-19, impact of COVID-19, and experience with COVID-19 vaccines.**
(DOCX)

## Author Contributions

**Conceptualization:** Obidimma Ezezika, Toluwalope Adedugbe, Chris Thompson.

**Data curation:** Toluwalope Adedugbe, Tatyana Graham, Bethelehem Girmay, Yanaminah Thullah.

**Formal analysis:** Tatyana Graham, Yanaminah Thullah.

**Funding acquisition:** Chris Thompson.

**Investigation:** Obidimma Ezezika, Isaac Jonas.

**Methodology:** Obidimma Ezezika, Isaac Jonas, Tatyana Graham, Bethelehem Girmay.

**Project administration:** Obidimma Ezezika, Toluwalope Adedugbe, Chris Thompson.

**Resources:** Obidimma Ezezika.

**Supervision:** Obidimma Ezezika, Toluwalope Adedugbe.

**Validation:** Meron Mengistu, Tatyana Graham.

**Writing – original draft:** Obidimma Ezezika, Meron Mengistu.

**Writing – review & editing:** Obidimma Ezezika, Meron Mengistu.

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
