## [Decision Letter · Decision Letter 0]

13 Nov 2023

PONE-D-23-27592Resilience Throughout COVID-19: Unmasking the realities of COVID-19 and vaccination facilitators, barriers, and attitudes Among Black CanadiansPLOS ONE

Dear Dr. Ezezika,

Thank you for submitting your manuscript to PLOS ONE. After careful consideration, we feel that it has merit but does not fully meet PLOS ONE’s publication criteria as it currently stands. Therefore, we invite you to submit a revised version of the manuscript that addresses the points raised during the review process.

It was indeed a pleasure to be a part of the review process for this work. I am delighted to recommend the manuscript for publication with the minor changes that have been incorporated. I believe that this work will make a valuable addition to the journal's content and contribute to the advancement of knowledge in our field.<o:p></o:p>

We look forward to receiving your revised manuscript.

Kind regards,

Udoka Okpalauwaekwe, MD, MPH, PhD

Academic Editor

PLOS ONE

Additional Editor Comments:

Your submission has completed the peer review process, and I have received feedback from the reviewers. I am pleased to inform you that the reviewers have found merit in your work and believe it holds potential for publication in our journal. However, they have also provided some comments and suggested minor revisions to enhance the clarity and quality of the manuscript.

The key areas highlighted by the reviewers for revision are listed in the reviewer comments.

We believe that addressing these minor comments will not only address the reviewers' concerns but also enhance the overall quality and impact of your manuscript.

Reviewers' comments:

Reviewer's Responses to Questions

**Comments to the Author**

1. Is the manuscript technically sound, and do the data support the conclusions?

Reviewer #1: Yes

Reviewer #2: Yes

2. Has the statistical analysis been performed appropriately and rigorously? 

Reviewer #1: Yes

Reviewer #2: Yes

3. Have the authors made all data underlying the findings in their manuscript fully available?

Reviewer #1: Yes

Reviewer #2: Yes

4. Is the manuscript presented in an intelligible fashion and written in standard English?

Reviewer #1: Yes

Reviewer #2: Yes

5. Review Comments to the Author

Reviewer #1: Thank you for the wonderful qualitative research.

I would like to suggest the following;

- In methods, please try to discuss the major variables, how you have selected participants (slightly detail) and how did you maintain the consent.

- In analysis, also mention how did you do quantitative analysis.

- Also try to add the implication of the study for addressing the issues.

- Better to include the specific recommendations for improving the status

- Please check and correct the write-up errors

Reviewer #2: 1. Authors have mentioned the data collection process [line 107-118]. It is said that there were two instruments to collect the data- FGD and Semi structured Interview. Authors highlighted that they had informed verbally and they mentioned that there two written consent forms. There is little confusion that whether they have taken the informed consent.

2. They have collected information only from Black Canadians and interpreted the results as they are discriminated and have less access to health care services. It would be better to compare with other privileged groups for this kind of inference. So, I would suggest more comparisons in the discussion section will enhance the power of the study.

Dr Khem B Karki

6. PLOS authors have the option to publish the peer review history of their article (what does this mean?). If published, this will include your full peer review and any attached files.

Reviewer #1: No

Reviewer #2: **Yes: **Khem B Karki

---

## [Author Response · Author response to Decision Letter 0]

19 Jan 2024

Reviewer response document has been attached.

---

## [Decision Letter · Decision Letter 1]

17 Mar 2024

PONE-D-23-27592R1Resilience Throughout COVID-19: Unmasking the realities of COVID-19 and vaccination facilitators, barriers, and attitudes Among Black CanadiansPLOS ONE

Dear Dr. Ezezika,

Thank you for submitting your manuscript to PLOS ONE. After careful consideration, we feel that it has merit but does not fully meet PLOS ONE’s publication criteria as it currently stands. Therefore, we invite you to submit a revised version of the manuscript that addresses the points raised during the review process.

We look forward to receiving your revised manuscript.

Kind regards,

Udoka Okpalauwaekwe, MD, MPH, PhD

Academic Editor

PLOS ONE

Journal Requirements:

**Additional Editor Comments:**

Please pay close  attention to the the reviewer comments in this revision.

Reviewers' comments:

Reviewer's Responses to Questions

**Comments to the Author**

1. If the authors have adequately addressed your comments raised in a previous round of review and you feel that this manuscript is now acceptable for publication, you may indicate that here to bypass the “Comments to the Author” section, enter your conflict of interest statement in the “Confidential to Editor” section, and submit your "Accept" recommendation.

Reviewer #1: All comments have been addressed

Reviewer #3: (No Response)

2. Is the manuscript technically sound, and do the data support the conclusions?

Reviewer #1: Yes

Reviewer #3: Yes

3. Has the statistical analysis been performed appropriately and rigorously? 

Reviewer #1: Yes

Reviewer #3: Yes

4. Have the authors made all data underlying the findings in their manuscript fully available?

Reviewer #1: Yes

Reviewer #3: Yes

5. Is the manuscript presented in an intelligible fashion and written in standard English?

Reviewer #1: Yes

Reviewer #3: Yes

6. Review Comments to the Author

Reviewer #1: (No Response)

Reviewer #3: This is a pertinent, comprehensive, and insightful manuscript. The authors should be commended for their work. There are just a few minor questions for clarifications. If addressed, the manuscript appears suitable for publication.

1. Lines 120 – 121 – “Interviews were conducted virtually over Zoom and lasted approximately 45 minutes”. Were incentives provided for participation?

2. Lines 128-129 – “ Focus groups were held virtually over Zoom and lasted approximately 45 minutes”. If both the semi-structured interview and focus groups were 45 minutes, how did they differ? Where the same questions used for each? Was an interview guide used?

3. Line 135 – “they met 20 times” – very thorough research.

4. Lines 168-170 – “For this study’s purposes, we limited our scope of analysis to sex (female or male) rather than gender, which encompasses a wide range of expressions of identities beyond biological attributes”. Good recognition. This could be seen as a limitation.

5. Line 482 – “Experiences with COVID-19 vaccines, with barriers such as policy & laws and limited access”. Consider removing the ampersand.

7. PLOS authors have the option to publish the peer review history of their article (what does this mean?). If published, this will include your full peer review and any attached files.

Reviewer #1: No

Reviewer #3: No

---

## [Decision Letter · Decision Letter 2]

21 May 2024

Resilience Throughout COVID-19: Unmasking the realities of COVID-19 and vaccination facilitators, barriers, and attitudes Among Black Canadians

PONE-D-23-27592R2

Dear Dr. Ezezika,

We’re pleased to inform you that your manuscript has been judged scientifically suitable for publication and will be formally accepted for publication once it meets all outstanding technical requirements.

Kind regards,

Udoka Okpalauwaekwe, MD, MPH, PhD

Academic Editor

PLOS ONE

Additional Editor Comments (optional):

Reviewers' comments:

Reviewer's Responses to Questions

**Comments to the Author**

1. If the authors have adequately addressed your comments raised in a previous round of review and you feel that this manuscript is now acceptable for publication, you may indicate that here to bypass the “Comments to the Author” section, enter your conflict of interest statement in the “Confidential to Editor” section, and submit your "Accept" recommendation.

Reviewer #3: All comments have been addressed

2. Is the manuscript technically sound, and do the data support the conclusions?

Reviewer #3: Yes

3. Has the statistical analysis been performed appropriately and rigorously? 

Reviewer #3: Yes

4. Have the authors made all data underlying the findings in their manuscript fully available?

Reviewer #3: Yes

5. Is the manuscript presented in an intelligible fashion and written in standard English?

Reviewer #3: Yes

6. Review Comments to the Author

Reviewer #3: The authors have sufficiently addressed the reviewer comments. The clarifications have enhanced the paper.

7. PLOS authors have the option to publish the peer review history of their article (what does this mean?). If published, this will include your full peer review and any attached files.

Reviewer #3: No

---

## [Editor Report · Acceptance letter]

14 Jun 2024

PONE-D-23-27592R2 

PLOS ONE

Dear Dr. Ezezika, 

I'm pleased to inform you that your manuscript has been deemed suitable for publication in PLOS ONE. Congratulations! Your manuscript is now being handed over to our production team.

Kind regards, 

on behalf of

Dr. Udoka Okpalauwaekwe 

Academic Editor

PLOS ONE